# Fourier Transform Infrared Spectroscopy Based Complementary Diagnosis Tool for Autism Spectrum Disorder in Children and Adolescents

**DOI:** 10.3390/molecules25092079

**Published:** 2020-04-29

**Authors:** Gulce Ogruc Ildiz, Sevgi Bayari, Ahmet Karadag, Ersin Kaygisiz, Rui Fausto

**Affiliations:** 1Department of Physics, Faculty of Sciences and Letters, Istanbul Kultur University, 34158 Istanbul, Turke; 2Department of Chemistry, CQC, University of Coimbra, P-3004-535 Coimbra, Portugal; rfausto@ci.uc.pt; 3Department of Physics Engineering, Hacettepe University, 06800 Ankara, Turkey; bayari@hacettepe.edu.tr; 4Department of Geological Engineering, Istanbul University-Cerrahpasa, 34320 Istanbul, Turkey; ersinkygsz@gmail.com; 5Department of Chemistry, King Fahd University of Petroleum and Minerals, Dhahran 34463, Saudi Arabia

**Keywords:** autism spectrum disorder, FTIR spectroscopy, chemometrics

## Abstract

Autism spectrum disorder (ASD) is a neurodevelopmental disorder that begins early in life and continues lifelong with strong personal and societal implications. It affects about 1%–2% of the children population in the world. The absence of auxiliary methods that can complement the clinical evaluation of ASD increases the probability of false identification of the disorder, especially in the case of very young children. In this study, analytical models for auxiliary diagnosis of ASD in children and adolescents, based on the analysis of patients’ blood serum ATR-FTIR (Attenuated Total Reflectance-Fourier Transform Infrared) spectra, were developed. The models use chemometrics (either Principal Component Analysis (PCA) or Partial Least Squares Discriminant Analysis (PLS-DA)) methods, with the infrared spectra being the *X*-predictor variables. The two developed models exhibit excellent classification performance for samples of ASD individuals vs. healthy controls. Interestingly, the simplest, unsupervised PCA-based model results to have a global performance identical to the more demanding, supervised (PLS-DA)-based model. The developed PCA-based model thus appears as the more economical alternative one for use in the clinical environment. Hierarchical clustering analysis performed on the full set of samples was also successful in discriminating the two groups.

## 1. Introduction

Autism spectrum disorder (ASD) is a neurodevelopmental disorder that begins early in life and continues lifelong. It affects about 1%–2% of the children population in the world. Metabolic diseases, as well as genetic, toxic and environmental factors are recognized as causes of ASD. Symptoms mainly appear as difficulties in social interaction and communication, as well as limited and repetitive patterns of behavior [1,2,3,4,5,6].

Current diagnosis of ASD is based only on the clinical evaluation of the behavioral signs and symptoms. The absence of auxiliary methods that can complement the clinical evaluation increases the probability of false identification of the disorder, especially in the case of very young children. On the other hand, studies have shown that ASD should be detected at the earliest possible age for the treatment to be effective [7,8].

There have been many attempts to find ASD biomarkers in genetics, neuroimaging, gene expression, and measures of the body’s metabolism. Body fluids are easy to collect, and their analysis is less expensive compared to neuroimaging and genome studies. Recent studies show that blood samples are one of the most promising targets to search for characteristic biomarkers for ASD, not only due to their easy accessibility but also because of their important biological information content on the health status of children with this disorder [1,4,9,10,11,12,13,14,15,16,17].

Many studies have been reported aiming to contribute to a better understanding of the underlying causes of ASD. Nevertheless, no biomarkers have yet been identified that can support the clinical diagnosis. Spectroscopic methods, in particular FTIR (Fourier Transform Infrared) spectroscopy, complemented with multivariate methods such as PCA (Principal Component Analysis) and PLS-DA (Partial Least Squares Discriminant Analysis), are becoming powerful tools for the analysis of biological samples, especially body fluids [18,19,20,21,22,23,24]. This approach joins the possibility offered by spectroscopy of using molecular information contained in the spectral data, with the analytical efficiency of multivariate statistical methods to process that information. As a whole, the method is cheap, essentially non-destructive and it can be easily implemented in clinical environment. FTIR spectroscopy on blood samples has been used for the investigation of various diseases, including Parkinson’s and Alzheimer’s, but also different types of cancer and infections, among many others [19,21,23,25,26,27,28]. In the present study, we developed chemometrics models based on FTIR data, which can be used as complementary diagnostic tools of ASD in children and adolescents. Besides, specific spectral regions were identified that can act as biomarkers to help distinguishing autistic from healthy individuals.

## 2. Materials and Methods

### 2.1. Clinical Stage

#### 2.1.1. Patients and Control Group Selection

A total of 60 children and adolescents (30 confirmed ASD cases and 30 controls; Table 1) participated in this study, after consents were obtained from their parents. The ASD group members were chosen among patients that are under treatment in the Child and Adolescent Psychiatry outpatient Clinic of the Pamukkale University (Denizli, Turkey). The group consisted of 23 boys and 7 girls within the age 4–17, diagnosed with ASD according to Diagnostic and Statistical Manual of Mental Disorders [2] criteria. In addition to the diagnostic evaluation, the Childhood Autism Rating Scale (CARS) [29] was applied to evaluate the severity level of the disease. Individuals with other psychiatric disorder and those having a chronic medical comorbid condition were excluded from the study. The healthy control included 22 boys and 8 girls, aged 6 to 16, without any medical or psychiatric history. The study has been approved by the Ethics Committee of Pamukkale University, Faculty of Medicine (date: 12/05/2015; number: 07).

#### 2.1.2. Samples Preparation

Five milliliters of venous blood was taken from the antecubital vein of all participants in the study. The collected blood samples were allowed to clot and then centrifuged for 15 min at 1000 rpm in order to separate the serum from cellular material. The obtained serum samples were aliquoted and stored at −20 °C until the analysis.

### 2.2. Spectroscopic Stage

#### 2.2.1. Sample Measurements

ATR-FTIR spectra were recorded on a Perkin Elmer Spectrum One spectrometer equipped with a KBr beam splitter and a deuterated triglycine sulfate (DTGS) detector, combined with a diamond GladiATR accessory (Pike Technologies). Sixty-four scans, covering the 4000–450 cm^−1^ wavenumber range, were co-added to produce each spectrum. A spectral resolution of 4 cm^−1^ was used. For each blood serum sample, 5 spectra were obtained.

Before collecting each spectrum, the ATR crystal was first cleaned using sterile phosphate buffer followed by ethanol. Background was collected prior to each sample measurement. For the spectra collection, 1 µL of unfrozen blood serum samples were placed on the crystal surface and allowed to air dry (~12 min) at room temperature.

#### 2.2.2. Data Pre-processing

Before analysis, the FTIR spectra were pre-processed by performing baseline correction, and area normalization. No smoothing or any other additional pre-processing of the spectra was performed. For the analyses, the 3700–2400 and 1800–900 cm^−1^ spectral regions were chosen. The full set of spectra belonging to the totality of samples of the control (C) or ASD (A) groups (5 × 30 spectra for each group) were then subjected to PCA, using the Nonlinear Iterative Partial Least Squares (NIPALS) algorithm [30], in order to detect outliers. This procedure resulted in the elimination of 5 replicas in total, all belonging to the same sample of the control group (C30 sample), which was excluded from the dataset. The average spectrum for each sample was then obtained, as well as the global mean-spectrum for each group (C and A).

All data pre-processing was undertaken with the Unscrambler^TM^ CAMO software (Version 10.5) [31].

#### 2.2.3. Classification Models Development and Testing

The dataset used to develop and test the classification models included a total of 59 samples, 30 belonging to the ASD group (A) and 29 to the control group (C). The calibration set comprehended 29 samples (15 for the A group and 14 for the C group), while the test set was formed by 15 samples of each group, in a total of 30 samples. The samples used in the calibration and test sets were chosen randomly.

Two models were built for classification of the samples, one using the PCA method and the other the PLS-DA method [32]. For both models, internal full cross-validation was used during calibration. For predictions, all samples in the test set were used with the two developed models. The hierarchical clustering technique was also applied to the full set of samples, as a preliminary unsupervised test to check the similarity of the samples within each group and the dissimilarity between the two groups. The performed cluster analysis used the Ward’s method with squared Euclidean distances [33,34]. All chemometric analyses were done using the Unscrambler^TM^ CAMO software (Version 10.5) [31].

The prediction performance of the models was checked by calculating their sensitivity, specificity, precision, accuracy, and efficiency [35,36]). Sensitivity and specificity measure the ability of the model to correctly classify each class and to correctly identify the samples that do not belong to the modelled class, respectively, and are calculated according to: Sensitivity (%) = 100 × tp / (tp + fn); Specificity (%) = 100 × tn / (tn + fp), where fp and fn stand for false positive and false negative samples, respectively, and tp and tn stands for true positive and true negative samples. Precision (%) = 100 × tp / (tp + fp), measures of the quality of the positive predictions of the model. Efficiency and accuracy provide a single measure of the model performance, with efficiency combining the information given by the sensitivity and specificity analyses [efficiency (%) = 100 × (sensitivity + specificity) / 2], and accuracy measuring the proportion of correct classifications independent of the class [accuracy (%) = 100 × (correct classifications) / total samples].

## 3. Results and Discussion

### 3.1. Preliminary Data Analysis

Figure 1 shows the average IR spectra (area normalized) of the blood serum of the ASD and control groups. Table 2 presents the assignment of the bands, according to the literature [37,38,39,40,41,42,43]. The data seems to indicate that the blood serum of the ASD patients have an increase of protein total contents and a slight decrease of tyrosine compared to the control group, while the lipids total amount seems to be nearly identical. Though being only indicative, these results agree with the conclusions of Croonenberghs et al. [44], who reported an increased level of the total protein contents in the blood serum of children with ASD, Elbaz and co-workers [10], Tu, Chen and He [45] and Tirouvanziam et al. [46], who found a significant reduction in tyrosine, and Wiest and co-workers [47], who concluded that the lipid contents in the blood plasma of ASD children and the general population are identical. Besides, the dietary study by Levy et al. [48] also indicated that the protein intake is incremented in ASD children patients.

Figure 2 presents the results of hierarchical clustering analysis performed on the full set of samples, which was undertaken as a preliminary unsupervised similarity test. The samples belonging to the two groups appear clearly discriminated. Noteworthy, the dendrogram also clearly shows that the homogeneity within the control group is significantly higher compared to the ASD group, as expected considering that ASD represents a range of mental disorders of the neurodevelopmental type with different levels of severity (these include disorders previously classified separately and designated as autism and childhood disintegrative disorders, Asperger’s syndrome, and pervasive developmental disorder not otherwise specified (PDD-NOS) [2]).

### 3.2. Classification Models Development

As mentioned before, two models were built for classification of the samples, one based on the PCA method (PCAModel) and the other on the PLS-DA method [32] (PLSModel). For both models the calibration set included the same 29 samples (15 for the A group and 14 for the C group), and internal full cross-validation was used.

Figure 3 shows the PCAModel 2D-scores plot (PC-2 vs. PC-1), where it can be seen that the ASD samples are well discriminated from those belonging to the control group along PC-1. Together, PC-1 and PC-2 explain 98% of the data variation for the training set (92% and 6% variance for PC-1 and PC-2 respectively), with the same numbers for validation. The model was developed using five principal components, accounting for a total variance of 99% for the training set (validation: 98%).

The loadings for PC-1 are given in Figure 4, where they are compared with the difference between the IR spectrum obtained by subtracting the average spectrum of the ASD group from the average spectrum of the control group (see Figure 1 for original spectra). The similarity between the data allows to assign a clear meaning to PC-1. In addition, this similarity can also be correlated with the circumstance that a large amount of variation in the dataset (92%) is explained by the first principal component. The fact that PC-1 loadings are very much similar to the difference between the average spectra of the two groups (control and ASD) is also relevant because it clearly demonstrates that the achieved discrimination is doubtlessly related with the different nature of the samples. Furthermore, it also validates our main approach to the problem under study, which is the statement that the whole IR spectrum acts as a holistic fingerprint (or spectroscopic biomarker) of the disease, since the PC-1 loadings have significant values for practically all variables (frequency values).

It is also interesting to note that the samples distribution within each group along PC-1 is substantially different, with the samples belonging to the control group spawning along a small range of values and those belonging to the ASD group (A) spreading a much wider range. This result reflects the greater homogeneity of the samples in the control group (C) compared to the ASD group, thus closely following the trend observed in the hierarchical clustering analysis discussed in the previous section. Along PC-2 the distribution of samples does not differ very much from one group to the other, indicating that the variance explained by this principal component reflects general small differences in the spectra that are not related with the different nature of the two groups (A vs. C).

The 2D-scores plot (Factor-2 vs. Factor-1) for the PLSModel is shown in Figure 5. As it could be expected considering the results obtained using the unsupervised PCA-based model (PCAModel) described above, the supervised PLSModel discriminates well the ASD samples from the control ones. The results obtained with the two models are in fact very similar. In the PLSModel, in parallel to what was found for the PCAModel, the groups are discriminated along the axis explaining the largest fraction of variance (Factor-1, whose loadings are also identical to the difference between the average IR spectra of the control and ASD groups, like the PC-1 loadings in the PCAModel), while along Factor-2 the distribution of samples are rather identical. Additionally, as it was found for the PCAModel, along Factor-1 the samples belonging to the ASD group are dispersed along a much wider range of score values than those belonging to the control group. The reasons for the two last trends, common to the two developed models, have already been provided above.

In the PLSModel, the two latent variables explaining the largest amounts of variation (Factor-1 and Factor-2) account for 97% of the variation in *X* and 90% in *Y* for the training set (91% and 80% variance in *X* and *Y*, respectively, for Factor-1, and 6% and 10% for Factor-2), with similar numbers observed for validation (91% and 5% variance in *X,* and 78% and 12% variance in *Y* for Factor-1 and -2 respectively; the totals were 96% and 90% for *X* and *Y* variance, respectively). The model was developed using five latent variables, accounting for total *X* and *Y* variances of 99% and 98% for the training set (validation: 98% and 93%). The root-mean-square errors (RMSE) for training and validation are 0.10 and 0.14, respectively, which demonstrate the good quality of the regressions.

### 3.3. Predictions

Fifteen samples of each group (A and C) not used for calibration of the models were used as test set for predictions. The IR spectra of the test samples were pre-processed following the same steps as for the samples used in the calibration of the models. The results obtained for the predictions done using the two models are summarized in Figure 6, Figure 7 and Figure 8.

Both models were able to classify correctly all samples included in the test set, with no false positives or negatives (global accuracy, 100%), so that the calculated values for all parameters chosen to measure the prediction performance of the models (sensitivity, specificity, precision, accuracy, and efficiency [35,36], whose meaning was given in Section 2.2.3) are maximal (100%). Figure 6 and Figure 7 show the projections of the test samples on the scores plots of the models, while Figure 8 gives the PLSModel predicted *Y* values for the samples and deviations, the last being an indicator of how reliable the predicted values are [49].

For the PCAModel, the predicted class for the samples was assigned based on the inspection of their projection on the model scores plot (Figure 6). The samples were ascribed as belonging to the same group to which the samples corresponding to its nearest three neighbor points belong. In the case of the PLSModel, the class assignments were done by considering the predicted *Y* values for the samples, using as threshold value for class separation the half distance between the reference *Y* values (0.5). It shall be noticed that the predicted *Y* values for the ASD test samples show a larger dispersion around the reference value compared to the control test samples. This result is in consonance with the already mentioned smaller homogeneity between the samples belonging to the ASD patients in comparison with those of the control group.

The fact that the two models show a similar prediction performance is striking. In fact, the performance of the PCA-based model is so good that the model derived using the supervised PLS-DA method, which could *a priori* be expected to outperform the unsupervised PCA-based model, does not improve on this latter in a noticeable way. Under these circumstances, the simplest PCAModel appears to be the most convenient model for practical use.

The PCAModel was also checked in relation to its robustness. For that, we performed an additional PCA calculation using all 59 samples in the training set. The idea was to verify if including a larger number of samples in the training set would visibly change the description of the data by the model. The obtained scores plot is presented in Figure 9. In this PCA PC-1 and PC-2 explain 95% of the data variation for the training set (89% and 6% variance for PC-1 and PC-2 respectively), with equal numbers for validation. The five principal components used in the PCA account for a total variance of 98% both for training set and validation. Noteworthy, the scores plot of this PCA (Figure 9), with all 59 samples in the calibration set, closely matches the projection graph obtained for the model developed with only 29 samples in the calibration set (see Figure 6), a result that clearly shows the robustness of the PCAModel.

## 4. Conclusions

In this study we developed analytical models for auxiliary diagnosis of ASD in children and adolescents, based on the analysis of patients’ blood serum infrared spectra. The models use chemometric (either PCA or PLS-DA) methods, with the infrared spectra acting as the *X*-predictor variables. The two developed models exhibit excellent classification/prediction performance. Remarkably, the simplest, unsupervised PCA-based model results to have a performance identical to the more expensive, supervised (PLS-DA)-based model. The developed PCA-based model thus appears as the best, more economical alternative for use in the clinical environment.

It can be concluded that infrared spectrum of the blood serum can be used for the discrimination of ASD patients from healthy controls. Since it considers the whole spectroscopic information to achieve classification, this approach is conceptually more consistent than the putative alternative ones that aim to use spectroscopic data of complex biological materials to find specific molecular biomarkers for the disease.

The obtained results regarding the relative similarity of the samples within each one of the two studied groups (ASD and control groups), clearly showing the much greater dissimilarity between the samples belonging to the ASD group, are in accordance with the modern psychiatric concept of ASD being a general type of disorder which includes a spectrum of clinical manifestations which previously were considered different psychiatric illnesses.

The methodology proposed herein is reliable, fast, cheap, essentially non-invasive, and might be implemented easily in the clinical environment in order to help psychiatrists to establish with increased certainty the ASD diagnosis at an early stage of development of the illness.

## Figures and Tables

**Figure 1 molecules-25-02079-f001:**
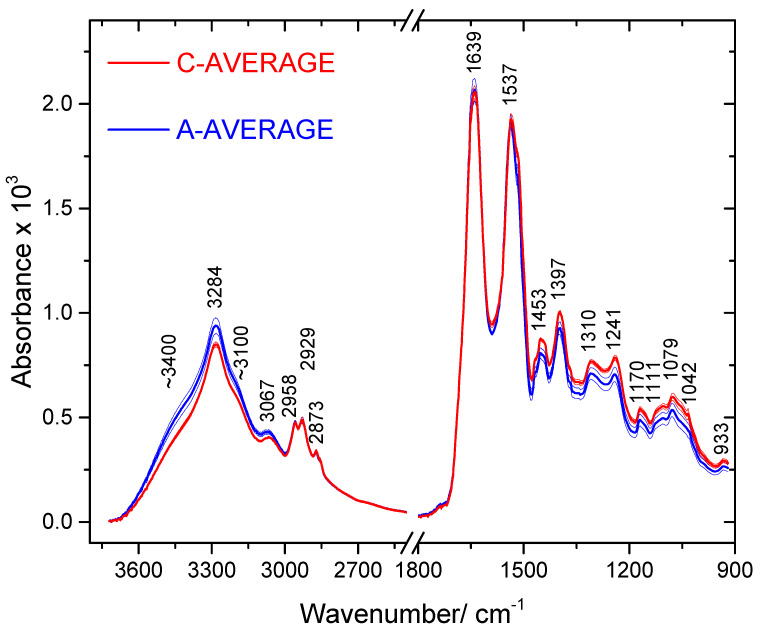
Average IR spectra of ASD (A-AVERAGE; thick blue line) and control (C-AVERAGE; thick red line) groups’ blood serum samples (3700–2400 and 1800–900 cm^−1^ regions). Thin lines account for the standard deviations.

**Figure 2 molecules-25-02079-f002:**
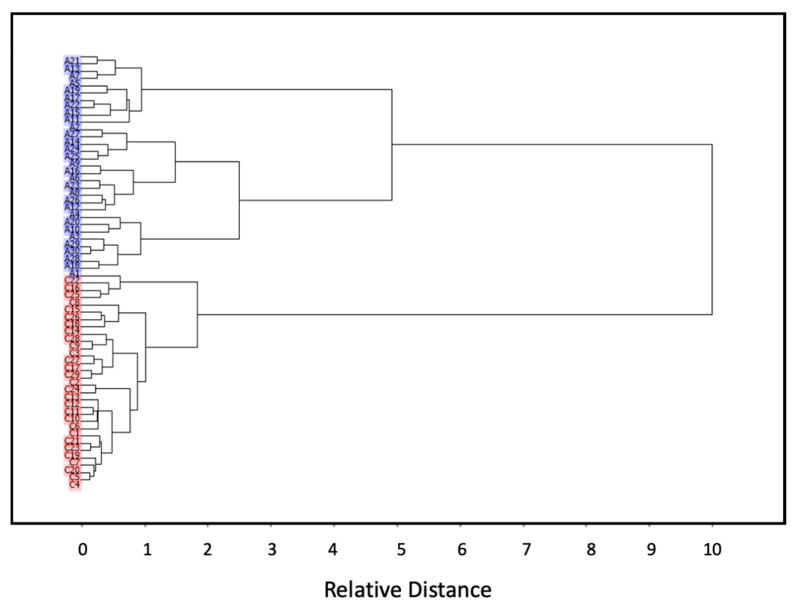
Cluster analysis of ASD (A; overlined in blue) and control (C; highlighted using the red color) groups’ blood serum IR spectra, according to the Ward’s method, using squared Euclidean distances.

**Figure 3 molecules-25-02079-f003:**
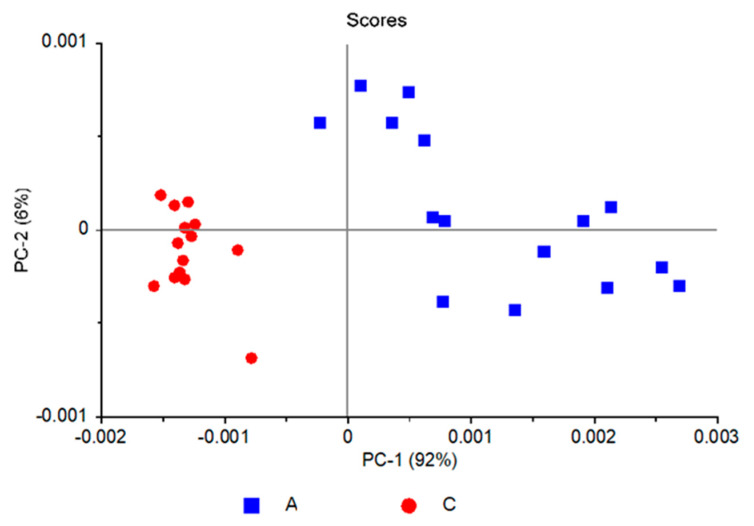
PCA scores plot (PC-2 vs. PC-1) for the PCAModel.

**Figure 4 molecules-25-02079-f004:**
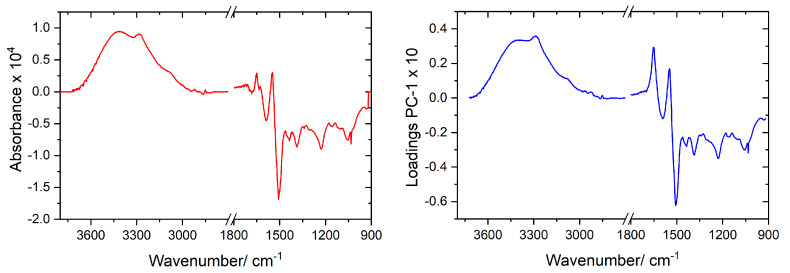
*Left panel*: Difference IR spectrum (average spectrum of ASD group (A) blood serum *minus* average spectrum of the control group (C) blood serum). *Right panel*: PC-1 loadings of PCAModel.

**Figure 5 molecules-25-02079-f005:**
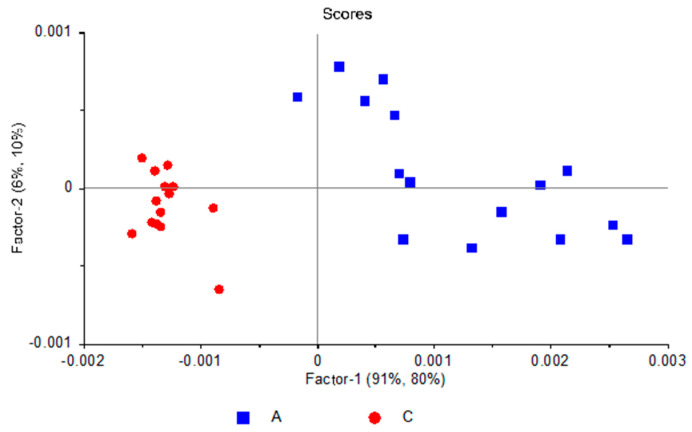
Scores plot (Factor-2 vs. Factor-1) for the PLSModel.

**Figure 6 molecules-25-02079-f006:**
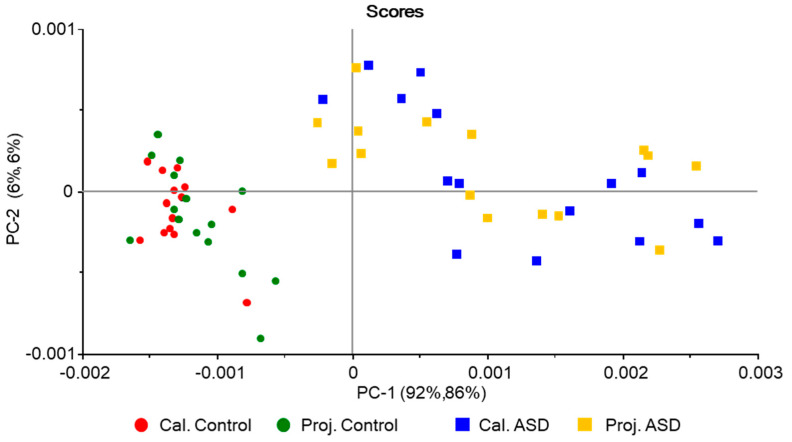
Projection scores plot (PC2 vs. PC-1) for the PCAModel.

**Figure 7 molecules-25-02079-f007:**
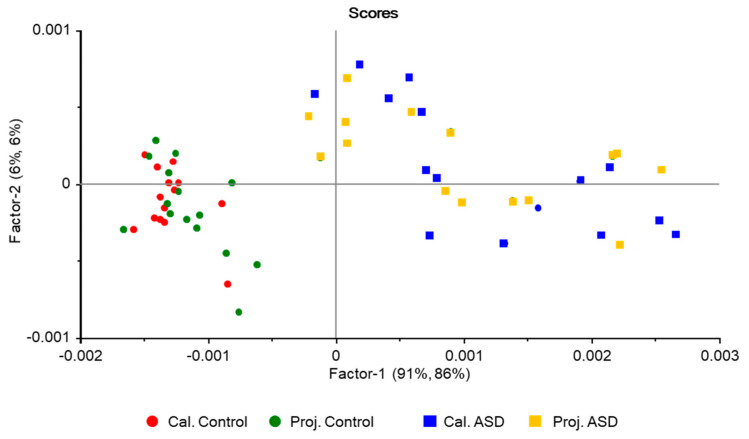
Projection scores plot (Factor-2 vs. Factor-1) for the PLSModel.

**Figure 8 molecules-25-02079-f008:**
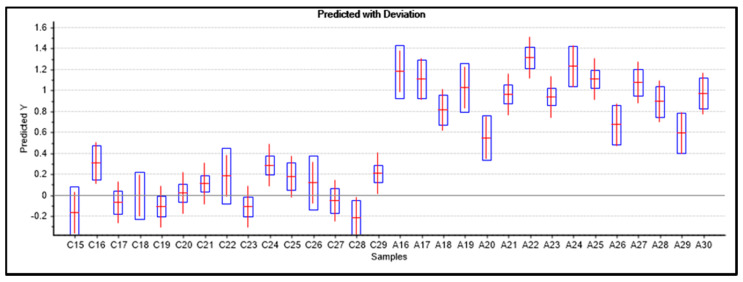
PLSModel predicted *Y* values for ASD (A group) and control (C group) test samples. The predicted values are indicated by the horizontal red lines, and the deviations by the blue boxes. In the model, samples belonging to control group define class 1 (value 0 for *Y*) and samples belonging to ASD patients define class 2 (value 1 for *Y*).

**Figure 9 molecules-25-02079-f009:**
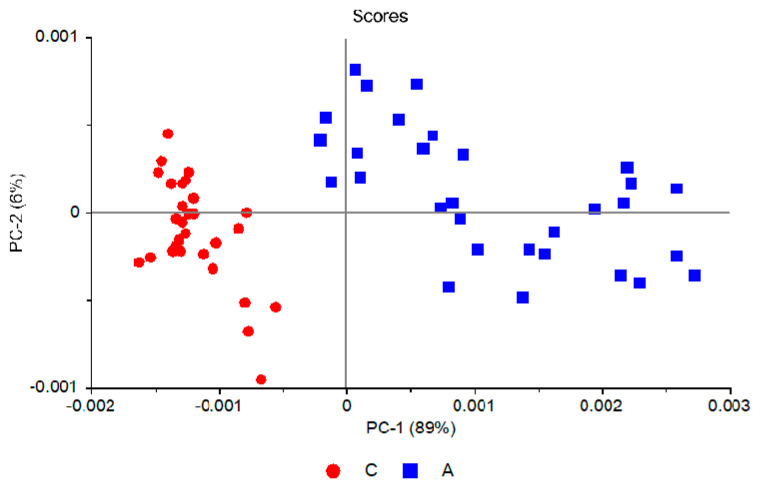
Scores plot (PC-2 vs. PC-1) for the PCA done using all 59 samples.

**Table 1 molecules-25-02079-t001:** Age and gender distribution in the samples, according to group (ASD or control) *^a^*.

ASD	Age	Sex	ASD	Age	Sex	Control	Age	Sex	Control	Age	Sex
**A1**	10	B	**A16**	9	B	**C1**	13	B	**C16**	10	B
**A2**	9	B	**A17**	7	B	**C2**	8	G	**C17**	11	B
**A3**	6	B	**A18**	4	G	**C3**	16	B	**C18**	10	B
**A4**	7	G	**A19**	8	B	**C4**	9	B	**C19**	8	B
**A5**	12	B	**A20**	5	B	**C5**	7	G	**C20**	8	B
**A6**	14	B	**A21**	5	B	**C6**	11	B	**C21**	16	B
**A7**	4	G	**A22**	5	B	**C7**	12	B	**C22**	10	B
**A8**	10	B	**A23**	7	B	**C8**	9	G	**C23**	10	B
**A9**	14	G	**A24**	17	B	**C9**	6	G	**C24**	11	B
**A10**	5	B	**A25**	7	B	**C10**	14	B	**C25**	8	B
**A11**	7	B	**A26**	7	B	**C11**	9	G	**C26**	12	B
**A12**	6	B	**A27**	13	G	**C12**	13	G	**C27**	8	G
**A13**	17	B	**A28**	8	G	**C13**	16	B	**C28**	8	B
**A14**	14	B	**A29**	4	B	**C14**	12	B	**C29**	10	B
**A15**	5	B	**A30**	13	G	**C15**	8	B	**C30**	7	G

*^a^* B, Boy; G, Girl. The first 15 members of the ASD group (A1-A15) and the 14 first members of the control group (C1-C14) were used to develop the models (calibration set); A16-A30 and C15-C29 (15 members of each group) were used to test the models; C30 was initially included in the calibration set, but was removed since it appeared as an outlier (see below). The numbering in the table was defined after the randomized split of the members of each group between the two sets (calibration and testing sets) and the preliminary investigation to exclude possible outliers.

**Table 2 molecules-25-02079-t002:** Assignments for the major bands in the FTIR spectrum of blood serum *^a^*.

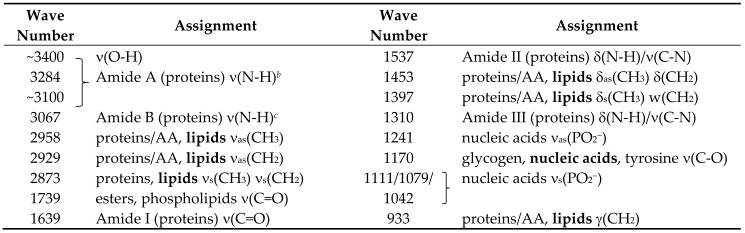

*^a^* The assignments are according to the literature [37,38,39,40,41,42,43]. Wavenumbers in cm^-1^. AA, amino acids; ν, stretching; δ, bending; w, wagging; γ, rocking; s, symmetric; as, anti-symmetric. Bold style in the assignment columns indicate the expected major contributor to the band. In the case of Amide II and III, the main coordinates contributing to the mode are indicated; the first mode corresponds to the anti-phase combination of these coordinates, while the second corresponds to the in-phase combination. *^b^* The high-wavenumber wing of the Amide A band is superimposed with the band originated in OH stretching vibrations, including those due to traces of water still present in the sample. *^c^* The Amide B band is partially due to N-H stretching vibrations of amide groups involved in strong intramolecular H-bonds, and partially a result of a Fermi resonance interaction between νNH and the first overtone of the Amide II vibration.

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
