# Peer review of "Fourier Transform Infrared Spectroscopy Based Complementary Diagnosis Tool for Autism Spectrum Disorder in Children and Adolescents"

_molecules, 2020, doi:10.3390/molecules25092079_

Round 1

Reviewer 1 Report

The original research article titled "FTIR Spectroscopy-based Complimentary Diagnosis Tool for Autism Spectrum Disorder in Children and Adolescents" submitted to the Molecules by Ildiz et al., was a quick and novel approach to diagnose the ASD patients. Authors utilized the analytical models for diagnosis of ASD collected from the blood serum of patients (children and adolescents) and tested on ATR-FTIR spectroscopy. Further, authors performed the PCA or PLS-DA - chemometrics, to classify ASD vs healthy controls. Authors concluded that PCA-based model could be an economical alternative to use in the clinical environment. Overall, this study is more relevant to clinical application of ASD patients hence, I would like to recommend to accept for publication in Molecules, after addressing the comments below.

Comments:

  1. Line #31, Expand in first instance of PCA or PLS-DA in abstract.
  2. Line #54, Rewrite this sentence "On the other hand, several studies have shown that in order the treatment to be effective, ASD must be detected at the earliest age possible [7,8]".
  3. Line #65, PCA (Principal Component Analysis) and PLS-DA (Partial Least Squares Discriminate Analysis), are becoming powerful tools for the analysis of biological samples.. why this is powerful method? Please explain it little more detail.
  4. Line #88, Sample preparation was not explained adequately, please specify how cellular materials were collected and in what form it was kept on the FTIR machine for analysis with what e.g., KBR etc.?
  5. Line #108, Principal Component Analysis (PCA) and line #121 Partial Least Squares Discriminant Analysis (PLS-DA) -redundancy it was already abbreviated in the introduction section.
  6. Line #153, Please print with mean +/- SD data for Fig 1. [ASD (A-AVERAGE; blue line) and control (C-AVERAGE; red line)]
  7. Line #205, Fig 4 legend in correctly printed "Figure 4. Difference IR spectrum (average spectrum of ASD group (A) blood serum minus average spectrum of the control group (C) blood serum (top) and PC-1 loadings of PCA Model (bottom)". it should be    left and right instead of top and bottom.
  8. Line #241, Change the Table 2 title. "Confusion Tables and statistical model performance indicators for the PCA Model and LS Model"
  9. Line #319, "It can be concluded that the infrared spectrum of the blood serum works as a sensitive probe for the disorder, acting as a holistic spectroscopic biomarker for the illnesses". I am afraid this could be overly stated, rephrased it.

Author Response

Answers to the reviewers and changes made to the manuscript FTIR Spectroscopy-based Complimentary Diagnosis Tool for Autism Spectrum Disorder in Children and Adolescents

(Manuscript ID: molecules-744686)

Below, the reviewers´ comments are in black. Our answers in blue.

Reviewer 1:

Comments and Suggestions for Authors

The original research article titled "FTIR Spectroscopy-based Complimentary Diagnosis Tool for Autism Spectrum Disorder in Children and Adolescents" submitted to the Molecules by Ildiz et al., was a quick and novel approach to diagnose the ASD patients. Authors utilized the analytical models for diagnosis of ASD collected from the blood serum of patients (children and adolescents) and tested on ATR-FTIR spectroscopy. Further, authors performed the PCA or PLS-DA - chemometrics, to classify ASD vs healthy controls. Authors concluded that PCA-based model could be an economical alternative to use in the clinical environment. Overall, this study is more relevant to clinical application of ASD patients hence, I would like to recommend to accept for publication in Molecules, after addressing the comments below.

 We thank the reviewer for his/her appreciation of our work.

Comments:

Point 1: Line #31, Expand in first instance of PCA or PLS-DA in abstract.

Response 1: The meaning of PCA and PLS-DA has been provided in the revised Abstract.

Point 2: Line #54, Rewrite this sentence "On the other hand, several studies have shown that in order the treatment to be effective, ASD must be detected at the earliest age possible [7,8]".

Response 2: The sentence has been rewritten as follows. “On the other hand, studies have shown that ASD should be detected at the earliest possible age for the treatment to be effective.”

Point 3: Line #65, PCA (Principal Component Analysis) and PLS-DA (Partial Least Squares Discriminant Analysis), are becoming powerful tools for the analysis of biological samples. why this is powerful method? Please explain it little more detail.

Response 3: A fragment of text has been including providing the necessary details (page 2): “This approach joins the possibility offered by spectroscopy of using molecular information contained in the spectral data, with the analytical efficiency of multivariate statistical methods to process that information. As a whole, the method is cheap, essentially non-destructive and can in general be easily implemented in clinical environment.”

Point 4: Line #88, Sample preparation was not explained adequately, please specify how cellular materials were collected and in what form it was kept on the FTIR machine for analysis with what e.g., KBR etc.?

Response 4: The requested information was already provided in the original manuscript. As explained in the “Samples Preparation” Section (Section 2.1.2):

5 mL of venous blood was taken from the antecubital vein of all participants in the study. The collected blood samples were allowed to clot and then centrifuged for 15 minutes at 1,000 rpm in order to separate the serum from cellular material. The obtained serum samples were aliquoted and stored at -20 °C until the analysis.”

Then, as explained in the second paragraph of the “Sample Measurements” Section (Section 2.2.1):

“The ATR crystal was first cleaned using sterile phosphate buffer followed by ethanol. Background was collected prior to each sample measurement. For the spectra collection, 1 µL of unfrozen blood serum samples were placed on the crystal surface and allowed to air dry (~12 minutes) at room temperature.”

Point 5: #108, Principal Component Analysis (PCA) and line #121 Partial Least Squares Discriminant Analysis (PLS-DA) -redundancy it was already abbreviated in the introduction section.

Response 5: Principal Component Analysis and Partial Least Squares Discriminant Analysis have been abbreviated as PCA and PLS-DA, respectively, in the “Introduction” Section, as requested by the reviewer.

Point 6: Line #153, Please print with mean +/- SD data for Fig 1. [ASD (A-AVERAGE; blue line) and control (C-AVERAGE; red line)]

Response 6: Figure 1 was changed to represent the average spectra +/- the standard deviations.

Point 7: #205, Fig 4 legend incorrectly printed "Figure 4. Difference IR spectrum (average spectrum of ASD group (A) blood serum minus average spectrum of the control group (C) blood serum (top) and PC-1 loadings of PCA Model (bottom)". it should be left and right instead of top and bottom.

Response 7: Legend of Figure 4 has been corrected. It now reads: “Figure 4. Difference IR spectrum (average spectrum of ASD group (A) blood serum minus average spectrum of the control group (C) blood serum (left) and PC-1 loadings of PCA Model (right)".

Point 8: Line #241, Change the Table 2 title. "Confusion Tables and statistical model performance indicators for the PCA Model and LS Model"

Response 8: The legend of Table 2 has been changed as requested to: "Confusion Tables and sensitivity, specificity, precision and efficiency values calculated for the calibrated PCA and PLS-DA models”.

Point 9: Line #319, "It can be concluded that the infrared spectrum of the blood serum works as a sensitive probe for the disorder, acting as a holistic spectroscopic biomarker for the illnesses". I am afraid this could be overly stated, rephrased it.

Response 9: The sentence has been reformulated: “It can be concluded that infrared spectrum of the blood serum can be used for the discrimination of ASD patients from healthy controls.”

Reviewer 2 Report

Comments on manuscript number 744686

The manuscript entitled “FTIR Spectroscopy-based Complimentary Diagnosis Tool for Autism Spectrum Disorder in Children and Adolescents” is an important contribution in the area, mostly because the number of spectroscopic investigations on body fluids is limited. Indeed, ASD diagnosis based only on clinical analyses may lead to flawed conclusions and so it greatly needs to be complemented by other characterization tools. Despite the relevance of the work, there are several issues that need to be addressed before its acceptance.   

  1. In my opinion, “Introduction” is short and I would expect to see information on other characterization spectroscopic techniques. For instance, reference [18] also employs UV-Vis analyses in the study of blood samples. Moreover, Chromatography is another important tool used to monitor body fluid of autistic children (please read Biomedical Chromatography 31 (2017) e3951). In this sense, the authors could then compare the techniques and emphasize the importance of vibrational spectroscopy and chemometric methods.
  2. In “Sample Measurements” the text needs to be improved and I imagine that the instrument has been placed in a box containing dry air flux. If no, please explain. Also, replace “cm-1” for “cm-1” at this point and in others along the text.
  3. In “Data Pre-Processing” the authors should inform which band has been used to normalize spectra and the reasons for its choice.
  4. Details on the internal full cross-validation, which has been used during calibration, could be included either in “Classification Models Development and Testing” or in Supplementary Material (SM).
  5. In “Results and Discussion” I recommend including the wavenumbers on each band in Figs. 1 and 4. By the way, there are two shoulders on the 3284 cm-1 band in Fig. 1, which were not assigned, but they become prominent features in the difference IR spectrum (Fig. 4). Please add a discussion on these signals to the text.

Although the interpretation regarding Fig.1 is in agreement with other studies (first paragraph), it is strongly dependent on the normalization employed and band assignment. Detail on the former was already requested while the second needs update (Table 1). Actually, the band at 1310 cm-1 corresponds to the νCN mode of amide, even though the authors inform that such vibration is mixed with δNH and both contribute to the 1537 cm-1 band. Regardless of this, the interpretation based on the behavior of these signals differs from the original one in terms of the increase in the total protein amount of ASD patients, as also supported by Fig.4. Hence, the authors should revisit the related papers and improve the discussion. Also, designations such as amide “A” and “B” are not clear.

Caption of Figure 4 is not right. Please change “top” for “left” and “bottom” for “right” at the present form. The information in the frame should be removed.

In “Predictions” the IR spectra of the test samples should be included in SM.

It is hard to accept the data reported in Table 2. To be honest, the error associated to each parameter should be calculated and included.

  1. Information in “Conclusions” could be somehow attenuated, mainly the last paragraph.
  2. English language could be corrected in some sentences (e.g. lines 62, 122, 137, 187, 188, etc…).
  3. In line 81 the term “5th edition” should be taken.
  4. Reference [35] is “Adv. Protein Chem.” and not “J. Adv. Protein Chem.”

Author Response

Answers to the reviewers and changes made to the manuscript FTIR Spectroscopy-based Complimentary Diagnosis Tool for Autism Spectrum Disorder in Children and Adolescents

(Manuscript ID: molecules-744686)

Below, the reviewers´ comments are in black. Our answers in blue.

Reviewer 2:

Comments and Suggestions for Authors

Comments on manuscript number 744686

The manuscript entitled “FTIR Spectroscopy-based Complimentary Diagnosis Tool for Autism Spectrum Disorder in Children and Adolescents” is an important contribution in the area, mostly because the number of spectroscopic investigations on body fluids is limited. Indeed, ASD diagnosis based only on clinical analyses may lead to flawed conclusions and so it greatly needs to be complemented by other characterization tools. Despite the relevance of the work, there are several issues that need to be addressed before its acceptance.   

We thank the reviewer for his/her appreciation of our work.

Point 1: In my opinion, “Introduction” is short and I would expect to see information on other characterization spectroscopic techniques. For instance, reference [18] also employs UV-Vis analyses in the study of blood samples. Moreover, Chromatography is another important tool used to monitor body fluid of autistic children (please read Biomedical Chromatography 31 (2017) e3951). In this sense, the authors could then compare the techniques and emphasize the importance of vibrational spectroscopy and chemometric methods.

Response 1: We have added some additional information regarding the advantages of the techniques we used (vibrational spectroscopy and chemometrics) in the “Introduction” section. We are of opinion that the references we give in the Introduction of our manuscript complement adequately the information we explicitly provide.

Point 2: In “Sample Measurements” the text needs to be improved and I imagine that the instrument has been placed in a box containing dry air flux. If no, please explain. Also, replace “cm-1” for “cm-1” at this point and in others along the text.

Response 2: The experiments were carried out by using ATR. In ATR, the attenuated total reflectance uses a property of total internal reflection called the evanescent wave. A beam of infrared light is passed through the ATR, which reflects it at least once off the internal surface in contact with the sample. This forms an evanescent wave which extends into the sample. The beam is then collected by a detector as it exits the crystal. In the case of a solid sample (like those studied in the present investigation), the sample is pressed into direct contact with the crystal, so that atmospheric air does not distort the results. This is in fact a well-known fact, so that in our opinion it does not require to be mentioned in the article. Of course we know that infrared spectroscopic measurements performed in transmittance mode (for example) require appropriate purging by a flux of dried, CO2 removed air (or a flux of N2).

Point 3: In “Data Pre-Processing” the authors should inform which band has been used to normalize spectra and the reasons for its choice.

Response 3: We suppose the something escaped here to the attention of the reviewer. Indeed, as stated in the manuscript (Section 2.2.2 Data Pre-processing”), the spectra were area normalized, which means that all the area defined by the whole spectrum is used for the normalization of each spectrum. This method is well-known and the reasons for using it is obvious: we need to keep the total intensity constant for all spectra used for the analysis (what is relevant in our study is the relationship between variables, and not the absolute magnitude of the response).

Point 4: Details on the internal full cross-validation, which has been used during calibration, could be included either in “Classification Models Development and Testing” or in Supplementary Material (SM).

Response 4: The details were in fact already indicated in the original version of the manuscript. Please see second paragraph on page 7, and also second paragraph on page 10.

Point 5: In “Results and Discussion” I recommend including the wavenumbers on each band in Figs. 1 and 4. By the way, there are two shoulders on the 3284 cm-1 band in Fig. 1, which were not assigned, but they become prominent features in the difference IR spectrum (Fig. 4). Please add a discussion on these signals to the text.

Response 5: Wavenumbers of bands’ maxima have been added to Figure 1. As usually, amide A band (which is mostly originated in amide N-H stretching vibrations) shows several maxima. The different maxima are related essentially with the fact that the amide N-H groups participate in H-bonds of different strength, whose precise origin cannot be determined, considering the complexity of the system under study. The high-wavenumber wing of the Amide A band is superimposed with the band originated in OH stretching vibrations, including those due to traces of water still present in the sample. We have now indicated also the wavenumbers of the shoulders of the Amide A main band in Table 1, and provided more details on the assignments in the footnotes of the same table. Because the detailed assignment of the spectra is not essential for the present study we prefer to keep the discussion about the assignments short. Detailed information can be find in the cited literature.

Point 6: Although the interpretation regarding Fig.1 is in agreement with other studies (first paragraph), it is strongly dependent on the normalization employed and band assignment. Detail on the former was already requested while the second needs update (Table 1). Actually, the band at 1310 cm-1corresponds to the νCN mode of amide, even though the authors inform that such vibration is mixed with δNH and both contribute to the 1537 cm-1 band. Regardless of this, the interpretation based on the behavior of these signals differs from the original one in terms of the increase in the total protein amount of ASD patients, as also supported by Fig.4. Hence, the authors should revisit the related papers and improve the discussion. Also, designations such as amide “A” and “B” are not clear.

Response 6: In point 3 above, we have already answered the question about normalization. The answer regarding band assignments has also already been partially addressed above (point 6). Amide I, II, III, A and B are the usual notations of bands for proteins. In any case, we have provided a description of all these modes in the revised version of the manuscript (description of Amide III and B were missing in the original version of Table 1 and have been added; in particular, we have included now in the table the description of Amide III protein band in terms of main contributor oscillators (amide dN-H and nC-N), as requested). Regarding the discussion about the amount of protein content in the ASD blood serum compared to healthy people: we tried to be very much cautious regarding this point in our manuscript, considering the results only “indicative”. Our approach to the problem we tackled is stressed in the “Conclusions” Section: “It can be concluded that infrared spectrum of the blood serum can be used for the discrimination of ASD patients from healthy controls. Since it considers the whole spectroscopic information to achieve classification, this approach is conceptually more consistent than the putative alternative ones that aim to use spectroscopic data of complex biological materials to find specific molecular biomarkers for the disease.”

Point 7: Caption of Figure 4 is not right. Please change “top” for “left” and “bottom” for “right” at the present form. The information in the frame should be removed.

Response 7: The caption of Figure 4 has been corrected and the information in the frame removed.

Point 8: In “Predictions” the IR spectra of the test samples should be included in SM.

Response 8: This corresponds to 5x30 (150) spectra. For consistency, the remaining 150 should also be included. We do not see any advantage in introducing all this material in the Supplementary Information.

Point 9: It is hard to accept the data reported in Table 2. To be honest, the error associated to each parameter should be calculated and included.

Response 9: The numbers provided in the table are counts. So, there are no associated errors. One count is one count, not “one count plus/minus something”.

Point 10: Information in “Conclusions” could be somehow attenuated, mainly the last paragraph.

Response 10: We have rephrased somehow the conclusion, in particular the second paragraph (as also requested by reviewer #1), in order to make it less striking.

Point 11: English language could be corrected in some sentences (e.g. lines 62, 122, 137, 187, 188, etc…).

Response 11: We have corrected the English mistakes in the indicated sentences and checked carefully all the manuscript regarding language.

Point 12: In line 81 the term “5th edition” should be taken.

Response 12: The term “5th edition” has been taken from the text.

Point 13: Reference [35] is “Adv. Protein Chem.” and not “J. Adv. Protein Chem.”

Response 13: Reference [35] has been corrected.

Reviewer 3 Report

The manuscript describes the use of ATR-FTIR spectroscopy of patient blood serum samples, coupled with multivariate methods, for the diagnosis of Autism Spectrum Disorder in children and adolescents. The study is well conceived and presented, and the results are of significance. There are however a number of issues which should be addressed to improve the quality of the presentation.

(i) In general, acronyms should not be used in the title (FTIR) and should be explained if used in the abstract (PCA, PLSDA)

(ii) Abstract -in what way is PLSDA "more demanding"?

(iii) 2.2.2. Data Pre-processing - How exactly was the baseline correction performed?

To be a supercritical "devil's advocate", it may be possible that the differentiation observed is due to a different background on the two datasets, which is been incompletely removed from one dataset. Comparing the spectra of Figure 1, almost the entire ASD spectrum is high on the high wavenumber side, and low on the Low wavenumber side.A difference for example in protein content should be manifest on both sides.

As a definitive check, the authors could compare the 2nd derivative spectra of the Raw, Uncorrected spectra.

(iv) PCA is normally not a classification method (unless coupled, for example with LDA). How exactly was the PCA model constructed?

(v) The hierarchical clustering technique should be mentioned in the abstract

Author Response

Answers to the reviewers and changes made to the manuscript FTIR Spectroscopy-based Complimentary Diagnosis Tool for Autism Spectrum Disorder in Children and Adolescents

(Manuscript ID: molecules-744686)

Below, the reviewers´ comments are in black. Our answers in blue.

Reviewer 3:

Comments and Suggestions for Authors

The manuscript describes the use of ATR-FTIR spectroscopy of patient blood serum samples, coupled with multivariate methods, for the diagnosis of Autism Spectrum Disorder in children and adolescents. The study is well conceived and presented, and the results are of significance. There are however a number of issues which should be addressed to improve the quality of the presentation.

 We thank the reviewer for his/her appreciation of our work.

Point 1: In general, acronyms should not be used in the title (FTIR) and should be explained if used in the abstract (PCA, PLSDA).

Response 1:  Title of the manuscript is changed as "Fourier Transform Infrared Spectroscopy Based Complimentary Diagnosis Tool for Autism Spectrum Disorder in Children and Adolescents" and PCA and PLS-DA are explained in the abstract.

Point 2: Abstract -in what way is PLSDA "more demanding"?

Response 2: PLS-DA is a supervised model where we shall define the groups a priori, while PCA is an unsupervised model, for which no information about the nature of the samples is required in advance. In this regard, PLS-DA is more demanded than PCA in terms of information required to be usable. But the PLS-DA method is also more demanding in terms of computational resources. When one reduces features using PCA, the idea is to maximize the variance of the features itself, such as the loss of information is greatly reduced; this is done by simple reorientation of the axes used to define the variables. On the other side, PLS uses the annotated label to maximize inter-class variance; that means it takes into account the classes and try to reduce the dimensions while maximizing the separation of classes, which is ideal in a classification scenario but mathematically more complex (since this requires a regression instead of simply reorienting the axes). To provide this sort of explanation in an Abstract (which has even a limited number of words imposed by the journal) is impossible, but we also think it is not essential, so that we would like to keep with the original version of the abstract that provides the most important clue regarding this issue: the unsupervised vs. supervised nature of the discussed methods.

Point 3: 2.2.2. Data Pre-processing - How exactly was the baseline correction performed? To be a supercritical "devil's advocate", it may be possible that the differentiation observed is due to a different background on the two datasets, which is been incompletely removed from one dataset. Comparing the spectra of Figure 1, almost the entire ASD spectrum is high on the high wavenumber side, and low on the Low wavenumber side. A difference for example in protein content should be manifest on both sides. As a definitive check, the authors could compare the 2nd derivative spectra of the Raw, Uncorrected spectra.

Response 3: Baseline correction was done automatically using the built-in algorithm of unscramble for baseline offset correction. All samples (both control and ASD) were processed at once. One also shall refer that the collection of spectra (all using the same collection parameters) was made choosing the samples in a random way, in order to eliminate any artifact due to the sequence used for obtaining the spectra. We followed the suggestion of the reviewer and perform the PCA using the 2nd derivative spectra of the raw, uncorrected spectra to check once more the effect of background. The PC2 vs. PC1 scores plot is shown below and, as it can be seen, discrimination between the two groups is also achieved in this case (though, as it could be expected, the results are not as good as using the original spectra subjected to the simple pre-processing described in the manuscript, mostly because derivatives always amplify noise).

Point 4: PCA is normally not a classification method (unless coupled, for example with LDA). How exactly was the PCA model constructed?

Response 4:  The reviewer touches an interesting point. Yes, in general PCA is not per se a classification method. However, in the present case, the PCA discrimination of the two sets (control and ASD) is good enough to allow direct use of the scores plot to classify the samples via projection. This is clearly shown in Figure 6 and it is pointed out in the text on pages 7 (bottom) and 9 (first paragraph) where the method used to classify the samples using the PCAModel is described.

Point 5: The hierarchical clustering technique should be mentioned in the abstract. 

Response 5:  A reference to the hierarchical clustering analysis has been introduced in the abstract, as requested.

Round 2

Reviewer 2 Report

Comments on manuscript number 744686_v2

The authors have partially answered to this reviewer’s comments so that important issues still need to be clarified before the acceptance of the manuscript.   

  1. [Data Pre-Processing] Normalization of spectra is a very well-known task, indeed. However, the normalized integrated intensities (areas) are obtained toward a band, which is often the most intense one in the spectrum and must be insensitive to environment changes. Hence, please inform which band has been used to normalize the whole spectrum and if it is in agreement with the aforementioned criteria.
  2. [Results and Discussion] The wavenumbers on each band should be included not only in Fig. 1, but also in Fig. 4, as recommended by this reviewer. Although Table 1 has been improved, the band at 1739 cm-1 cannot be seen in Fig. 1 and the rocking vibration is often denoted by r and not γ, as shown in the footnote.

I agree with the authors about the “indicative” results extracted from Fig.1. Nevertheless, I insist in saying that the corresponding text needs to be rewritten in order to avoid misunderstanding the reader. Again, the interpretation based on the signals at 1537 and 1310 cm-1 differs from the one based on the νNH region in terms of the increase in the total protein amount of ASD patients. It is worth stressing this latter region is strongly influenced by water absorption, as the authors also state. In addition, the presence of water could also affect the intensity of the 1639 cm-1 band and provide a result similar to the one observed at the νNH region.

Please remove the information in Fig. 1 (in the frame).

[Predictions] I disagree with the authors and kindly ask them to include selected IR spectra in the Supplementary Material. This will allow better comparison between the data.

[Table 2] The authors’ response is in full disagreement with the scope of the Analytical section and so they need to get a better way to present the data.

  1. [Conclusions] I still think that the last paragraph is a bit exaggerated.
  2. English language could be corrected in some sentences (e.g. line 134).

Author Response

-----------------